# Nanopore sequencing data analysis using Microsoft Azure cloud computing service

**Linh Truong** [1,2]*, **Felipe Ayora** [3], **Lloyd D'Orsogna** [1,2], **Patricia Martinez**[1,2], **Dianne De Santis**[1,2]

**1** Department of Clinical Immunology, PathWest, Perth, Australia, **2** UWA Medical School, University of Western Australia, Perth, Australia, **3** Research and Advanced Computing, BizData, Wellington, New Zealand

* huynh.truong@health.wa.gov.au

## Abstract

Genetic information provides insights into the exome, genome, epigenetics and structural organisation of the organism. Given the enormous amount of genetic information, scientists are able to perform mammoth tasks to improve the standard of health care such as determining genetic influences on outcome of allogeneic transplantation. Cloud based computing has increasingly become a key choice for many scientists, engineers and institutions as it offers on-demand network access and users can conveniently rent rather than buy all required computing resources. With the positive advancements of cloud computing and nanopore sequencing data output, we were motivated to develop an automated and scalable analysis pipeline utilizing cloud infrastructure in Microsoft Azure to accelerate HLA genotyping service and improve the efficiency of the workflow at lower cost. In this study, we describe (i) the selection process for suitable virtual machine sizes for computing resources to balance between the best performance versus cost effectiveness; (ii) the building of Docker containers to include all tools in the cloud computational environment; (iii) the comparison of HLA genotype concordance between the in-house manual method and the automated cloud-based pipeline to assess data accuracy. In conclusion, the Microsoft Azure cloud based data analysis pipeline was shown to meet all the key imperatives for performance, cost, usability, simplicity and accuracy. Importantly, the pipeline allows for the ongoing maintenance and testing of version changes before implementation. This pipeline is suitable for the data analysis from MinION sequencing platform and could be adopted for other data analysis application processes.

## Introduction

Advancement in genomic analysis has contributed to several scientific breakthroughs over the last decade. Genetic information provides insights into the exome, genome, epigenetics and structural organisation of the organism. Given the enormous amount of information, scientists are able to perform mammoth tasks to improve the standard of health care such as determining genetic influences on outcome of allogeneic transplantation, identifying disease-causing

**Data Availability Statement:** Yes - sample data are fully available without restriction. All FASTQ files from this study are available from the Dryad database (https://doi.org/10.5061/dryad.x0k6djhp4).

**Funding:** This study was funded by Innovation Grant (Microsoft Australia). The authors (LT, FA, DDS) were granted $35,000 AUD for the development of the automatic pipeline in Microsoft Azure server. The sponsor (Microsoft Australia) played no role in the study design, data analysis or preparation of the manuscript.

**Competing interests:** The authors have declared that no competing interests exist.

genes, predicting candidature proteins for vaccine development within an unprecedented speed as seen in the development of the messenger RNA (mRNA) vaccine against severe acute respiratory syndrome coronavirus 2 (SARS-CoV-2), tracking community spread of new viral outbreaks using 'genomic fingerprinting', and many more [1].

Nanopore sequencing from Oxford Nanopore Technologies (ONT) became commercially available in 2016 and has since embarked a new era of long-read single molecule technology. The ultra-long reads from ONT assist the visualization of genetic information in real-time without the need of a reference sequence as currently required by other short-read technologies. Nanopore sequencing also requires a lower capital footprint for instrumentation and reagents; however, it also creates new computational problems. Converting raw sequencing data to scientific results requires computational power, coordinated automation and storage capacity [2].

High number of compute resources is required to deploy the ONT data analysis pipeline consisting of base-calling, a process that converts raw electrical signal to nucleotide sequence, demultiplexing samples in the data set, and filtering based on predetermined quality thresholds. Additionally, the series of specialised tools used during the analysis often require access to GPU (Graphics Processing Unit) and CPU (Central Processing Unit) resources, as well as different operating systems, tools, applications and prerequisites. This complex combination of hardware and software resources forces the laboratory technicians to follow a time-consuming manual process including switching between different operating systems and being on stand-by for the duration of long analysis steps before starting the next one. Therefore, coordinated automation is desirable to increase the throughput without sacrificing the valuable time of our scientists and technicians.

The computational resources that are needed to analyse nanopore sequencing data in a timely-manner and to enable long-term storage has also outgrown the infrastructure owned by a single laboratory such as the Department of Clinical Immunology (DCI) located within PathWest Laboratory Medicine Western Australia (PathWest). Specifically, the raw data of one single ONT run could be as large as 250 Gigabytes (result of a R10.3 flow cell in 16 hours sequencing run) and the processed data could be up 20 Gigabytes in our experience. That means ONT workflow would produce approximately 35 Tetrabytes annually in our centre, while our current testing policy requires permanent storage of patient's clinical data, both raw and processed data. Therefore, the cost of purchasing physical storage hardware to keep up with the ONT output would become a burden to the centre's operating budget.

The Department of Clinical Immunology is the sole state provider for HLA genotyping, providing HLA gene characterization on patients awaiting bone marrow transplantation as well as potential unrelated donors recruited to the Australian Bone Marrow Donor Registry, which is then linked to worldwide registries. When a patient needs a stem cell transplant, their HLA genotype is compared with all potential donors on the worldwide registries. Having more enlisted local donors with high-resolution HLA typing increases the chance of finding the best-matched local donor for patients. This means that local patients can be transplanted faster and at lower cost compared with using an overseas donor. Additionally, high-resolution HLA typing of donors at the point of recruitment provides more information about the donor's immunogenetic make-up to the clinician and transplant team, therefore eliminating potential unknown mismatches during the donor selection process, and allows patients to proceed to transplant quickly which can directly influence the patient's survival outcome. Historically, all HLA type data from the MinION platform were processed manually using physical computing resources available on-site at Fiona Stanley Hospital. In order to increase the throughput and decrease the processing time, it was desirable to seek an affordable and stream-lined method for genetic analysis.

Cloud based computing has increasingly become a key choice for many scientists, engineers and institutions as it offers on-demand network access and users can conveniently rent rather than buy all required computing resources. Cloud providers refer to major commercial services such as Amazon Web Services (AWS), Google Cloud Platform or Microsoft Azure. All cloud providers offer elasticity, convenience and scalability depending on specific demand of individual workflow. Furthermore, they ensure the security and safety to store encrypted data for long-term while maintaining the utmost confidentiality of genetic information [3]. With the positive advancements of cloud computing and ONT data output, we were motivated to develop an automated and scalable analysis pipeline utilizing cloud infrastructure in Microsoft Azure to accelerate HLA genotyping service and improve the efficiency of the workflow at lower cost. Microsoft Azure was chosen for this study as PathWest has an active subscription with Microsoft as part of the IT arrangement for the health network. Therefore, as a default we had access to Microsoft services such as Microsoft 365 and Azure platform. In this study, we describe (i) the selection process of suitable virtual machine sizes for computing resources to balance between the best performance versus cost effectiveness; (ii) the building of Docker containers to include all tools in the cloud computational environment; (iii) the comparison of HLA genotype concordance between the in-house manual method and the automated cloud-based pipeline to assess data accuracy.

## Methods and results

### Workflow overview

The raw data from Oxford Nanopore Technologies (ONT) MinION platform was processed using multiple online bioinformatics tools. Changes in the voltage and raw signalling data was acquired by MinKNOW software (ONT) as the sequencing run progressed, converted and stored as FAST5 file format for downstream processing. Basecalling of raw data was performed using Guppy v4.0.14, a data processing toolkit provided by ONT, which provided a basecaller tool based on a recurrent neural network algorithm that converted raw nanopore signals into nucleotide sequences and wrote the results in FASTQ file format.

All FASTQ files were then de-multiplexed by the indexes-sorting tool in Guppy v4.0.14, which assigned reads according to the ligated molecular barcode into separate folders. The results were in multiple files containing reads from the same individual. The sequences corresponding to the molecular barcoded sequences were also trimmed by the Guppy Basecaller at the completion of de-multiplexing process. The individual FASTQ reads were then further filtered by size, a minimum length of 2 kbases, and quality, minimum Q-score of 7 by NanoFilt software tool [4] (https://github.com/wdecoster/nanofilt). The smallest HLA amplicon within this study amplicon pool was 3 kbases in size, therefore, any read shorter than 2 kbases was filtered to eliminate unbound primers, primer dimer or any potential non-specific products [5]. The Phred quality score or Q score is the most common metric used to assess the accuracy of sequencing technology. In earlier publications of ONT dataset, Q7 reads were considered as the benchmark for quality threshold of ONT sequencing, e.g. any reads with Q-score less than Q7 was defaulted into the "fail" bin and any reads with Q-score equal or higher than Q7 was sorted into the "past" bin. Therefore, read length of 2 kbases and minimum Q-score of Q7 were used for filtering parameter.

Finally, the quality of the sequencing run was monitored using MinIONQC [6] (https://github.com/roblanf/minion_qc). MinIONQC produced a sequencing summary outlining the data yield overtime, data total output, read quality histogram, read length histogram and Q-score obtained over time. The overview of each analysis step is shown in Fig 1. This workflow was then built into a pipeline of applications, all included within a single Docker container so

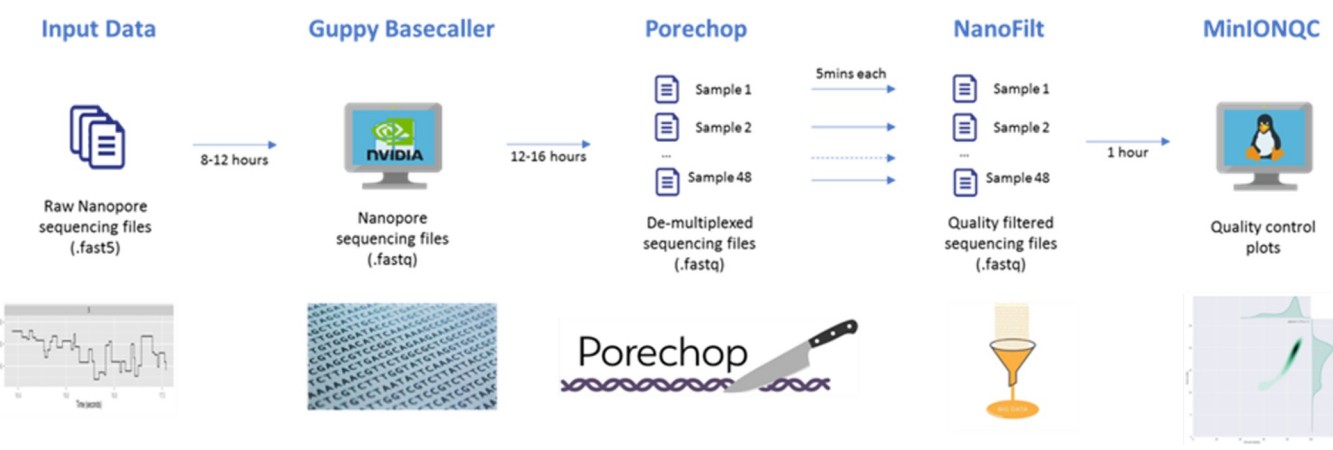

**Fig 1. Overview of the analytic workflow for ONT data.**

that they can be deployed quickly and scalable to on-demand cloud compute resources, and so that software versions and updates can be easily managed. The protocol described in this peer-reviewed article is published on protocols.io (dx.doi.org/10.17504/protocols.io.x54v9dj7pg3e/v1]dx.doi.org/10.17504/protocols.io.x54v9dj7pg3e/v1) and is included for printing purposes as S1 File.

## Building the pipeline in cloud computing

A set of predetermined criteria was applied during the building and deployment of the Azure cloud-based pipeline: (1) The pipeline would give preference to Platform as a Service (PaaS) and Software/Solution as a Service (SaaS) technologies, over Infrastructure as a Service (IaaS) technologies, in order to benefit the most from the cloud platform capabilities. (2) Sequencing runs can be processed in parallel, without having to queue them or change the pipeline. (3) The pipeline would match performance to cost, to achieve a balance that produces results in line with cost and analysis time expectations. (4) The pipeline would utilise the Loome plat-form (https://www.loomesoftware.com) for job orchestration, data movement and logging. To run jobs, Loome would deploy the pipeline using Docker containers running on GPU-based resources on Azure, and the resources must be automatically deleted when jobs have completed. When running, the automated genetic analysis pipeline should be accessible for monitoring, troubleshooting and alerting with detailed task execution history. (5) The speed to obtain the dataset using the Azure cloud-based analysis pipeline must be faster than the manual on-premises analysis pipeline. (6) The processing time must be less than 3 days and ideally no more than 12 hours to ensure that the overall assay turnaround time remains equivalent to the current assay turnaround of 5 days. (7) The upload of raw FAST5 files into the Azure cloud storage must be automated and progressive as they are generated by the MinION device. The download of analysed FASTQ result files back to Immunology (Fiona Stanley Hospital site) must also be automated and progressive. (8) The result datasets should be comparable between the Azure cloud analysis pipeline and manual on-premises analysis pipeline. The HLA geno-type results should be concordant between two workflows. (9) The nanopore sequencing data and analysed results cannot leave Australian data centres.

## Overview of architecture of the Azure cloud-workflow

The architecture of analysis pipeline on the Azure was designed as shown in Fig 2.

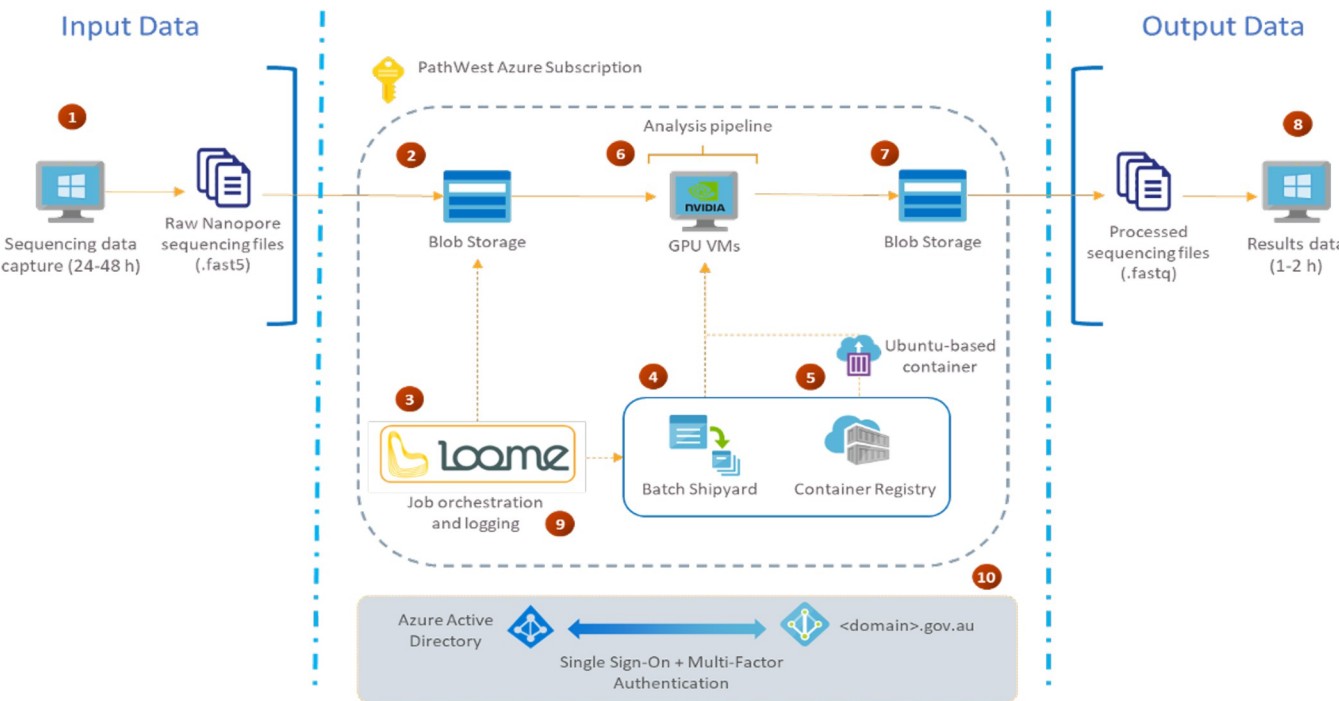

**Fig 2. The architecture of analysis pipeline on the Microsoft Azure.**

(1) The FAST5 sequencing files from MinKNOW acquisition software were exported to a default folder on the stand-alone computer progressively as they were generated. (2) The input files were automatically uploaded by Loome from the stand-alone computer into a container in a blob storage account, deployed within the PathWest Azure subscription. The files were uploaded using Transport Layer Security (TLS), and was encrypted at rest using 256-bit AES encryption. (3) The Loome agent running in the PathWest Azure subscription detected the presence of a new input dataset and triggered a new job to deploy the necessary resources, and to start the processing steps, using the Azure Batch service. (4) Azure Batch automatically deployed a GPU-enabled Virtual Machine (VM) for basecalling, de-multiplexing, quality trimming and QC overview. (5) As part of the job submission process, Loome communicated with Azure Batch which Docker container to use during each task in the job, and that container was pulled from the private Azure Container Registry and instantiated in each of the VMs that are managed by Azure Batch. (6) When each of the VMs was running, they copied the input data into their local disk for faster processing, run the analyses, and then copied the results back into blob storage so that the VMs could be deleted when processing had been completed. Loome in coordination with Azure Batch orchestrated these steps. (7) The results were stored in a blob storage account within the PathWest Azure subscription, ready to be downloaded. Loome detected the successful completion of all tasks in the job and sent an email to notify that the analysis had completed. (8) The analysed files in FASTQ format were then automatically downloaded from the blob storage account using Transport Layer Security (TLS), onto the stand-alone computer where downstream HLA analysis would be performed. (9) If troubleshooting was required, a user with administrative permission in Loome could log on using their existing enterprise account credentials, to review the detailed logs for each job and task. (10) Access to all Azure resources and to Loome was secured by Azure Active Directory, applying role-based access controls to existing Active Directory user accounts in the PathWest directory, and enforcing multi-factor authentication.

**Table 1. GPU VMs that were tested in the validation.**

| VM size | GPUs | Cost (hr) | Cost Per GPU | Finding | Runtime | Run Cost |
|---------|------|-----------|--------------|---------|---------|----------|
| NC24 v3 | 4 | $ 23.2531 | $ 5.8133 | Too costly | Not ran | |
| NC6 v3 | 1 | $ 5.8133 | $ 5.8133 | Best fit | 0:15:14 | $1.476 |
| NV24 | 4 | $ 8.7295 | $ 2.1824 | Incompatible GPU type | Not ran | |
| NV48 v3 | 4 | $ 8.7295 | $ 2.1824 | Incompatible GPU type | Not ran | |

(**Best fit**: identified to provide the best performance vs. cost results; **Too costly**: prices per GPU higher than comparable VM sizes; **Incompatible GPU type**: a GPU that supports NVIDIA CUDA drivers and complies with its licensing requirements is required to run Guppy analyses performantly. Only NC-series VMs currently provide this on Azure.

## Identifying optimal virtual machine

An analysis of different types of compute resources was performed to determine the best performance versus cost efficiency. This analysis also helped decide how to best containerise the applications, to match the resources required by them. A representative data subset was used for testing and the cost calculation analyses. A range of virtual machines (VMs) that were available in the Azure datacentres in Australia was evaluated. The available VMs for GPU applications included NC24 v3, NC6 v3, NV24 and NV48 v3. The VMs for CPU applications included D15v2, L32, G5, DS5 v2, F16s, DS32 v3, F72s v2, F32s v2, HC44, HB120 and HB60. The results from testing the pipeline applications, end to end, are shown in Tables 1 and 2.

The data used for the cost calculation analyses constituted only a representative subset of a complete MinION sequencing data (e.g. 50 Gigabytes of 150 Gigabytes of data were used for testing). For that reason, the runtime results shown in the Tables 1 and 2 were considerably lower than the runtime of a complete analysis. An extrapolation of 2x to 3x the total runtime of the cost calculation analyses was used as an approximation to estimate the runtime of a complete analysis. The key imperatives in selecting the suitable VM for GPUs and CPUs

**Table 2. CPU VMs that were tested in the validation.**

| VM size | vCPUS | Cost (hr) | Cost Per CPU core | Finding | Runtime | Run Cost |
|---------|-------|-----------|-------------------|---------|---------|----------|
| D15v2 | 20 | $ 2.7391 | $ 0.1370 | Low CPU count | Not ran | |
| L32 | 32 | $ 4.1080 | $ 0.1284 | Best fit | 3:57:06 | $16.23 |
| G5 | 32 | $ 14.0156 | $ 0.4380 | Too costly | Not ran | |
| DS5 v2 | 16 | $ 1.8467 | $ 0.1154 | Low CPU count | Not ran | |
| F16s | 16 | $ 1.4307 | $ 0.0894 | Long runtime | 5:03:10 | $7.23 |
| DS32 v3 | 32 | $ 2.7460 | $ 0.0858 | Hyperthreading | Not ran | |
| F72s v2 | 72 | $ 5.4865 | $ 0.0762 | Hyperthreading | 3:37:01 | $19.84 |
| F32s v2 | 32 | $ 2.4384 | $ 0.0762 | Hyperthreading | Not ran | |
| HC44 | 44 | $ 2.8270 | $ 0.0643 | Not in Australia | Not ran | |
| HB120 | 120 | $ 6.4256 | $ 0.0535 | Not in Australia | Not ran | |
| HB60 | 60 | $ 2.0348 | $ 0.0339 | Not in Australia | Not ran | |

(**Low CPU count**: during testing, Porechop was found to utilise up to 30 CPU cores to speed up processing. VMs with less than 30 CPU cores were determined to have too low a CPU count to achieve the faster results; **Best fit**: identified to provide the best performance vs. cost results; **Too costly**: prices per CPU core higher than comparable VM sizes; **Long runtime**: these VMs do not comply with the faster results solution; **Hyperthreading**: Hyperthreading splits physical CPU cores amongst running processes, which could cause a considerable decrease in performance, when compared to non-Hyperthreaded VMs, and running CPU-intensive applications; **Not in Australia**: one of the key requirements was to keep the sequencing data within Australian datacentres.)

included availability within Australian server, compatibility with the application, non-hyper-threading configuration and most importantly balance between cost and performance. For the GPU evaluation, both NC24 v3 and NC6 v3 VMs were suitable. The NC6 v3 consisted of one GPU while NC24 v3 offered four units, hence NC24 v3 would cost almost 4 times as much as NC6 v3 per hour of rent albeit it could potentially out-perform the NC6 v3 GPU by 4-folds. The analysis workflow on premise only involved one GPU, therefore the cost-benefit and comparable computing power to the manual process of NC6 v3 appeared superior to NC24 v3 (Table 1). Nc24 v3 was selected as the final candidate for GPU VM evaluation.

In the CPU VM evaluation, there were more options available within Australian Azure server without the hyperthreading possibility compared to GPU VM. Among five available CPU (D15v2, L32, G5, DS5v2 and F16s), L32 and G5 offered equivalent CPU power as the onsite computing unit, specifically 32 CPUs. It is important to note that G5 VM would cost approximately 3.5 times more than L32 machine, therefore G5 VM was not included in the testing and L32 VM was selected as the final candidate. In conclusion, the most optimal GPU-enabled VM was NC6 v3 with 1 NVIDIA Tesla V100 GPU, 6 Intel Xeon E5-2690 v4 (Broadwell) CPUs, 112 GB of RAM and P10 disk. The most compatible CPU-enabled VM was L32 with 32 Intel Xeon E5 v3 CPUs, 256 GB of RAM and P10 disk.

## Cost estimation for genetic analysis using Azure cloud-pipeline

After the best VM sizes were identified, it was possible to estimate the total cost per sample, as show in the Table 3 below. It is important to note that the usage of CPU-enabled VM was superseded with the implementation of the barcoding tool in Guppy and no longer included in the pipeline and cost-estimation below. The analysis cost per sample was estimated at $0.25 for a run of 48 samples, which was significantly lower compared to $5 per sample by manual on-premise analysis. The manual cost calculation was based on 3 hours of labour cost of a laboratory scientist to complete the execution.

The initial capital footprint to purchase physical computing infrastructure for the manual process cost approximately $3500 for the specification of Intel® Core™ i&-7700K CPU @ 4.20Ghz, 32 GB RAM, 64-bit operating system and GPU driver NVIDIA GTX 1080 Ti. However, this computer was for communal usage and not designated for TGS data analysis solely, therefore the onsite infrastructure and ongoing maintenance cost was not incorporated into the data analysis cost calculation. Similarly, PathWest holds an active subscription to the Microsoft Azure cloud server that is accessible to all departments within the organization. As the computing resource in Azure cloud was charged for the length of usage to perform the required task for current tenant, the licence or subscription fee to Azure was not included in this study. Overall, this validation was only based on the labour cost to perform analysis manually versus the cost to perform analysis by cloud computing for side-by-side cost comparison.

## Data analysis processing time evaluation

A comparison of analysis time between the manual analysis pipeline and the automated Azure cloud-based analysis pipeline was performed on MinION sequencing run 15_06_20_M1 with

**Table 3. Estimation of running cost per sample.**

| Compute | Time (hr) | Cost (hr) | Note |
|---|---|---|---|
| **GPU applications** | 2.12 | $ 5.6405 | NC6 v3 (1x V100 GPU, 112 GB RAM, 1x P10 disk) |
| **Cost per run** | | $ 11.96 | Includes VM start-up and deletion totalling 12 mins |
| **Cost per sample** | | $ 0.25 | 48 samples per run |

**Table 4. Comparison of the running time required to complete the analysis for a MinION run between the automatic pipeline in the Azure cloud and the manual process onsite.**

| ANALYSIS TIME THROUGH AUTOMATIC CPU AND GPU PIPELINE (AZURE CLOUD) | | | | | |
|---|---|---|---|---|---|
| **Run ID** | **Timeline** | **Date** | **Start** | **End** | **Duration** |
| 15_06_20_M1 | Guppy Basecalling (v3.6.1) | 16-06-2020 | 9:00 | 10:32 | 01:32 |
| | Porechop | 16-06-2020 | 10:32 | 17:57 | 07:25 |
| | NanoFilt, MinIONQC | 16-06-2020 | 17:57 | 18:38 | 00:41 |
| | | | | **Total** | **09:38** |
| **ANALYSIS TIME THROUGH AUTOMATIC GPU-ONLY PIPELINE (AZURE CLOUD)** | | | | | |
| **Run ID** | **Timeline** | **Date** | **Start** | **End** | **Duration** |
| 15_06_20_M1 | Guppy Basecalling (v4.0.14) | 05-11-2021 | 10:37 | 11:56 | 01:19 |
| | Guppy Barcoding (v4.0.14) | 05-11-2021 | 11:56 | 12:18 | 00:22 |
| | NanoFilt, MinIONQC | 05-11-2021 | 12:18 | 12:32 | 00:14 |
| | | | | **Total** | **01:55** |
| **ANALYSIS TIME THROUGH MANUAL PROCESS (ON PREMISES)** | | | | | |
| **Run ID** | **Timeline** | **Date** | **Start** | **End** | **Duration** |
| 15_06_20_M1 | Guppy Basecalling (v3.6.1) | 17-06-2020 | (17–06) 18:06 | (18–06) 02:24 | 08:18 |
| | Porechop | 18-06-2020 | 08:30 | 20:51 | 12:21 |
| | NanoFilt, MinIONQC | 19-06-2020 | 08:30 | 09:15 | 00:45 |
| | | | | **Total** | **21:24** |

16 hours-worth of data output (250 Gigabytes of data). The breakdown of running time is shown in Table 4.

Initially, when using a combination of CPU and GPU VMs and Porechop instead of Guppy for barcoding and demultiplexing, the full analysis of a representative MinION dataset using the Azure cloud infrastructure completed in 9 hours and 38 minutes compared to 21 hours and 24 minutes using the manual analysis pipeline on the DCI computer workstations. That represents a time-to-answer speed-up of 2.22x, or a reduction of analysis run time to approximately 45% of the manual process. It is also important to note that even though the total running time of the onsite analysis was ~ 21 hours, the full analysis took place over three days as each process was triggered manually and sequentially by an operator. If one step finished outside of the 8-hours routine working day, the next step was delayed until the next business day. Through automation & the flexibility of cloud architecture, it was possible to obtain the fully analysed FASTQ files from a complete MinION run within one working day without operator intervention.

After consolidation of the complete pipeline on GPU VMs by incorporating the barcoding and demultiplexing tool in Guppy which can take advantage of GPU resources, the full analysis of the same representative MinION data set on Azure completed in 1 hours and 55 minutes. That represents a time-to-answer speed-up of 11.17x, or a reduction of analysis run time to approximately 9% of the manual process.

## Genetic data output comparison

The study data set consisted of 48 representative samples from a well-characterized DNA panel selected for HLA antigens commonly found in Western Australian population. All genomic DNA was extracted from the peripheral white blood cells or B-cell transformed cell lines by QIAsymphony DNA Midi Kit (Qiagen, Germany) according to the vendor's protocol. The concentration and purity of extracted DNA were assessed by the optical density (OD) 260/280 ratio of 1.6–2.0. Furthermore, all samples had historical HLA high-resolution results available, obtained using the Ion Torrent platform [5].

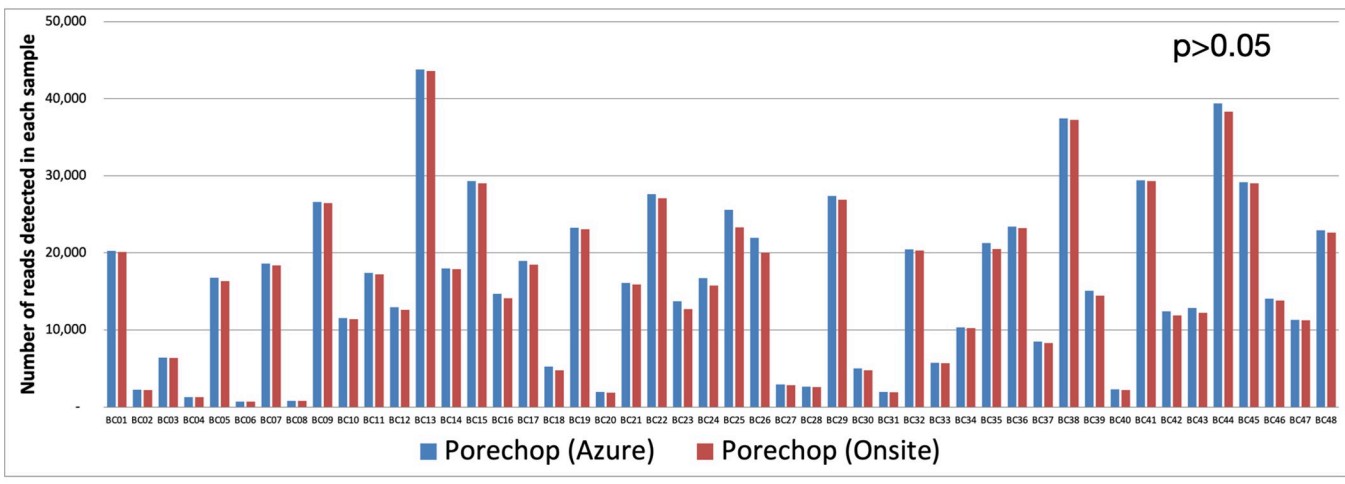

**Fig 3. The number of reads detected in each sample on a 48-samples run.** The blue colour bar represents the data obtained from Azure cloud analysis and the red colour bar depicts the data from manual analysis onsite.

The FASTQ files obtained from the manual and Azure analysis pipelines were examined. The number of reads in each FASTQ files from the Porechop output and NanoFilt output summaries were used for data yield comparison. The data output from the two analysis pipelines was strikingly comparable with p-value of 0.94 as shown in Fig 3. Therefore, the difference between two data sets was not statistically significant.

The demultiplexed output for each sample and the proportion of unclassified reads in the run were also comparable between two analytic pipelines (Fig 4). As expected, the output from two analysis pipelines had p-value of 1 and not statistically different. Ideally, the number of

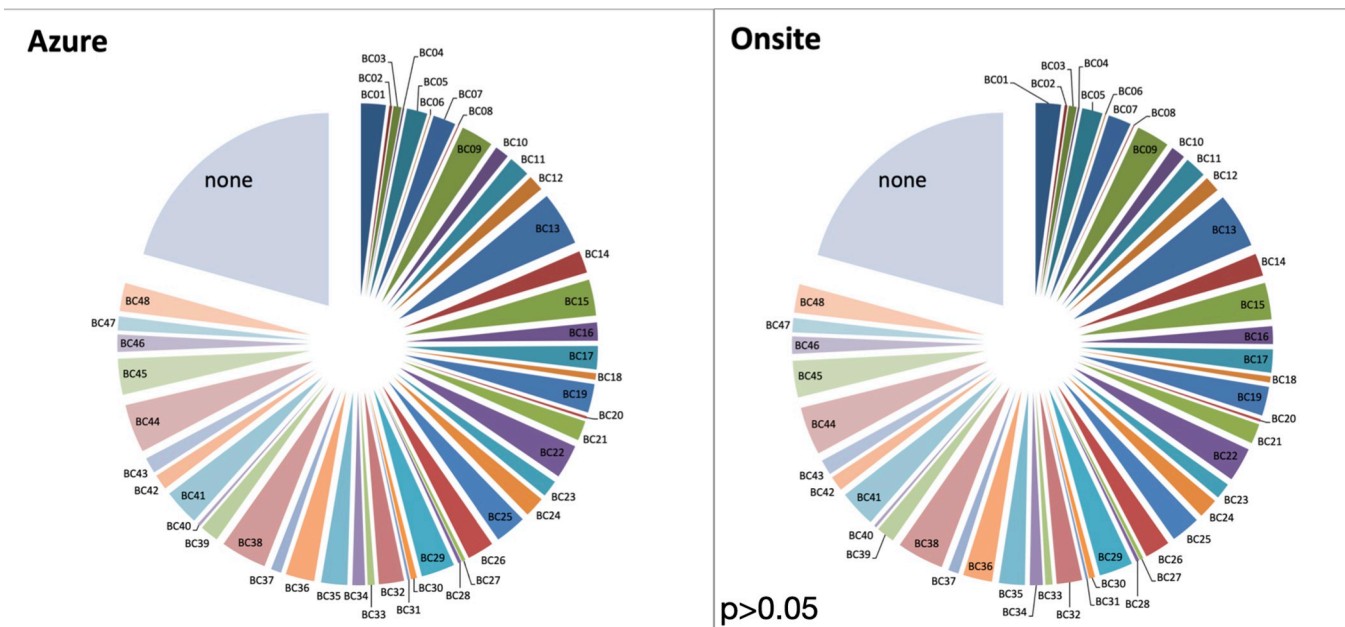

**Fig 4. The comparison of sample composition in a sequencing run between the automatic pipeline in the Azure cloud and the manual process onsite.**

**Fig 5. Analysis snapshot from NGSengine of sample #38 obtained from the automatic pipeline in Azure cloud platform (A panel) and the manual analysis workflow onsite (B panel).**

reads between the Azure and manual pipeline should be identical in each sample, however, minor differences in the output was acceptable only if the HLA genotype calls were concordant between the two pipelines. For example, sample 22 contained 27635 reads and 27097 reads using Azure and manual analysis, respectively. The gap between two pipelines was 538 reads, however, there was more than the minimum threshold of 3000 reads per sample for analysis obtained by both workflows. Most importantly, the HLA types were 100% concordant in sample 22.

HLA genotyping analysis using GenDx NGSengine (Utrecht, The Netherlands) software was performed on each sample in two data sets to determine data accuracy. There were very minor differences in the number of reads utilized by the HLA allele assignment software NGSengine to assign an HLA genotype between the data generated for each data set for the same sample. However, read depth for each gene analysed met the acceptance criteria of > 20 reads and there were no HLA genotype discrepancies in the data generated by each of the two pipelines. For instance, in sample #38, the number of reads matching to a HLA reference was 4112 reads using Azure analysis and 4139 reads using the manual onsite analysis as shown in the Fig 5A and 5B. The genotyping results were consistent and the presence of a novel polymorphism in an HLA-C*03 allele was identified in the data generated using the two analysis pipelines.

Overall, the data quantity and quality were comparable between the Azure cloud and manual on-premises analysis pipelines. There were no differences in the HLA genotype results obtained from either data analysis pipelines.

## Docker container overview

Docker containers were chosen for the nanopore sequencing pipeline since the Azure platform provides enhanced support for running containers at the right scale for the specialised hardware, such as hosts with GPUs and high-performance computing (HPC) clusters, both of which were required for this genomics pipeline. The dockerfile for GPU apps could be followed as:

```
# Base image from NVIDIA so that the GPU drivers and libraries are available to Guppy

FROM nvidia/cuda:11.3.0-cudnn8-runtime-ubuntu18.04.04

ARG GENERAL_DEPENDENCIES = "wget apt-transport-https software-properties-common"

ARG GUPPY_DEPENDENCIES = "lsb-release"

ARG NANOFILT_DEPENDENCIES = "python3-pip python3-pkg-resources dos2unix"

ARG MINIONQC_DEPENDENCIES = "r-base"

# Set non-interactive mode to override any user inputs requested by package installations

ENV DEBIAN_FRONTEND noninteractive

# Install dependencies

RUN apt-get update && \
    apt-get install—yes $GUPPY_DEPENDENCIES $MINIONQC_DEPENDENCIES $NANOFILT_DEPENDENCIES $GENERAL_DEPENDENCIES &&
    apt-get clean

 # Install AzCopy (data movement tool)

RUN mkdir -p /home/azcopy && cd /home/azcopy && \
    wget -O azcopy.tar.gz https://aka.ms/downloadazcopy-v10-linux && \
    tar -xf azcopy.tar.gz—strip = 1

ENV PATH = /home/azcopy:$PATH

 # Install NanoFilt

RUN pip3 install—upgrade pip

RUN pip3 install nanofilt

 # Install MinIONQC

 # Set the CRAN mirror as Perth (7th in the index = Curtin University's mirror) and then install dependency packages

RUN R -e "chooseCRANmirror(graphics = FALSE, ind = 6);install.packages(c('rlang', 'data.table', 'futile.logger', 'ggplot2', 'optparse', 'plyr', 'readr', 'reshape2', 'scales', 'viridis', 'yaml'))" &&
    export R_LIBS = "/usr/local/lib/R/site-library"

# Finally, download the R script for MinIONQC directly

RUN wget -O MinIONQC.R https://raw.githubusercontent.com/roblanf/minion_qc/master/MinIONQC.R

 # Install Guppy with dependencies
```

```
# *********************************************************************************

  # ***NOTE: The next line can be modified to create a container with a specific version
of Guppy (e.g., a new one for testing) ***

# *********************************************************************************

ARG GUPPY_VERSION = 4.0.14

# Requires setting DEBIAN_FRONTEND as non-interactive or keyboard-layout instal-
lation stalls the installation of Guppy

ARG DEBIAN_FRONTEND = noninteractive

ARG DEPENDENCY_PACKAGES = "libhdf5-100 libnorm1 libpgm-5.2–0 libsodium23
libzmq5 libhdf5-cpp-100 libboost-thread1.65.1 libboost-atomic1.65.1 libboost-
chrono1.65.1 libboost-date-time1.65.1 libboost-filesystem1.65.1 libboost-iostreams1.65.1
libboost-program-options1.65.1 libboost-regex1.65.1 libboost-system1.65.1 libboost-
log1.65.1"

RUN wget https://mirror.oxfordnanoportal.com/software/analysis/ont_guppy_
${GUPPY_VERSION}-1~bionic_amd64.deb &&

    apt-get install—yes $DEPENDENCY_PACKAGES &&

    dpkg -i *.deb

  # Install dateutils to for date and time calculations

RUN apt-get install -y dateutils

  # Install PowerShell Core

RUN wget -q https://packages.microsoft.com/config/ubuntu/18.04/packages-microsoft-
prod.deb &&

    dpkg -i packages-microsoft-prod.deb && \
    apt-get update && \
    add-apt-repository universe && \
    apt-get install -y powershell

  # Define Command or Entry Point

CMD ["/bin/bash"]
```

(Highlighted value: the value for this variable can be modified to build the container with a
different version of Guppy)

## Ethical approval

Written consent for genetic analysis (HLA genotyping in specific) was obtained for all samples
at the point of collection. For patient under 18 years old, the written consent was sought from
parents or guardians. Ethical approval for storage and biobanking of PBMC and DNA samples

at Fiona Stanley Hospital has been granted (RGS 0552) by Human Research Ethics Committee (PathWest, Department of Health). The main purpose of this study was to assess the analysis HLA genetic information using the cloud compute resources. No additional genetic information outside of the HLA would be unveiled from the genome of the study panel. All samples that were selected for this study had been de-identified for personal information, therefore the genomic data of these samples can be shared or published without jeopardising personal privacy and confidentiality. Additionally, other private information such as name, address, clinical history or treatment was not collected for the scope of this study. Lastly, only human samples were included in this research and no animal was involved.

## Discussion

Cloud computing server has been utilized ubiquitously in processing clinical trial data such as cancer treatment trial [7] and genetic studies [8]. To our knowledge, this study was the first in the world to leverage cloud computing to analyse TGS raw data for clinical HLA genotyping. The main goal of this study was to develop an automatic data analysis pipeline that streamlined the data flow from the MinION sequencing device to cloud computing and then back to the hospital network for downstream genomic analysis. This pipeline leveraged the scalability and flexibility of the cloud computing resources to produce the end-result of demultiplexed filtered FASTQ files at 11.17x times faster than the manual on-premises data analysis pipeline. Data analysis of MinION sequencing data took over 3 working days however with the implementation of the Azure cloud-based data analysis pipeline this was reduced to under 2 hours with the use of GPU-enabled VMs.

In the latest cloud pipeline, the total analysis cost was estimated at $11.96 per run or $0.25 per sample compared to $250 per run or $5 per sample if perform manually. Furthermore, the Azure cloud-based data analysis pipeline produced data quantity and quality comparable to that of the manual on-premises data analysis pipeline. There were no differences in the HLA genotype results obtained from both data analysis pipelines. Lastly, all data were stored in the data centre within Australia with utmost security. All samples were de-identified to adhere strict patient confidentiality.

This study showed that the most optimal GPU-enabled virtual machine (VM) was NC6 v3 with 1 GPU, 112 GB RAM and P10 disk. All GPU applications were stored in the Docker containers and managed by Loome and the Azure Batch service. The Loome agent communicated with Azure Batch and triggered the deployment of necessary computing resources. The Loome agent also orchestrated the movement of data from Blob Storage to VM and back to Blob Storage readily for transferring to the hospital stand-alone computer. The usage of Docker for containerisation would ensure the possibility of updating applications involved in the workflow such as Guppy, which has important updates approximately every 6 months.

It is important to note that the testing and optimization of this pipeline commenced in 2020, and the completed in 2021. The current cloud infrastructure and costing of computing resources remain as per the description in the method and result section. The analysis workflow has shown to be resilient over time as this pipeline with specific virtual machine (NC6 v3 with 1 NVIDIA Tesla V100 GPU, 6 Intel Xeon E5-2690 v4 (Broadwell) CPUs, 112 GB of RAM and P10 disk) is able to analyse the latest version of MinION flow cell (R10.4) and sequencing chemistry (Q20) using the same cost & computing set-up, both reagents were released in early 2022 (data not shown in this study). Furthermore, this pipeline is currently adopted for TGS raw data analysis for HLA typing in decease donor organ workup, which is time-sensitive and requires immediate turn-around-time.

A major imitation of this study is the dependency on availability of programmer and IT support in the building of Docker containers and orchestrating Loome agent, container

registry, VM and blob storage in Azure platform. One medical scientist from PathWest with entry-level skill in data science and one programmer from BizData/Microsoft had worked closely to build and optimize this cloud-based pipeline over the course of six months. The prolonged testing time was not due to the complexity of the pipeline or lack of manpower, the major hurdle was to obtain clearance to create a connection bridging the cloud server and the health network, while maintaining compliance with the cyber security and patient confidentiality policy of Fiona Stanley Hospital and Western Australian Department of Health.

In addition, IT support or a bioinformatician is required for the testing of application updates in the aforementioned Docker container prior implementation into production environment. Guppy base-caller algorithm is being upgrading periodically to improve the accuracy of the technology; therefore, it is important to keep up to date with the latest version of application for highest quality of sequencing data. The codes to build and maintain Docker image are open-access and made available in this study in order to assist the laboratories with limited access to programmer. However, in order to replicate this pipeline in a different laboratory, one would still require IT support and subscription to Microsoft Azure platform or cloud service provider of choice.

The common concern among medical scientists and clinicians with the usage of cloud server is the potential breach of patient's confidentiality and sensitive medical data such as HLA genotype results. The construction and deployment of this analysis pipeline in the Azure server had undergone the highest level of scrutiny and approval by the PathWest IT manager as well as network engineer manager at Fiona Stanley Hospital. The senior authors of this study provided a Deployment Architecture Design Document to the site manager outlining the requirements for network configuration, system configuration and the adherence to data security policy.

Specifically, the raw data sent to the Azure server do not contain patient's demographic data and consists only of electrical signals. The processed data of nucleotide sequences sent back to the hospital site is linked to a laboratory reference number. To identify patient data, one would require access to the PathWest Laboratory Information System which requires password access, to link the laboratory number to a patient's demographic data. Furthermore, the end-user would also require access to allele assignment program such as GenDx NGSengine to decipher nucleotide sequences to HLA types. Therefore, there are several checkpoints and multiple layers of security to warrant the utmost protection of patient's clinical data.

Overall, the Microsoft Azure cloud-based data analysis pipeline was shown to meet all the key imperatives for performance, cost, usability, simplicity and accuracy. Importantly, the pipeline allows for the on-going maintenance and testing of version changes before implementation. This pipeline is suitable for the data analysis from MinION sequencing platforms and could be adopted for other data analysis application processes.

## Supporting information

**S1 File.**
(PDF)

## Acknowledgments

We would like to thank Mr Johnny Gorea for his valuable input in the conceptualization of this study. We are also grateful to all the colleagues at the Department of Clinical Immunology, PathWest for their technical assistance and troubleshooting.

## Author Contributions

**Conceptualization:** Linh Truong, Felipe Ayora, Dianne De Santis.

**Data curation:** Linh Truong.

**Software:** Felipe Ayora.

**Supervision:** Lloyd D'Orsogna, Patricia Martinez, Dianne De Santis.

**Writing – original draft:** Linh Truong, Felipe Ayora.

**Writing – review & editing:** Lloyd D'Orsogna, Patricia Martinez, Dianne De Santis.

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
