## [Decision Letter · Decision Letter 0]

5 Sep 2022

PONE-D-22-07900Nanopore sequencing data analysis using Microsoft Azure cloud computing servicePLOS ONE

Dear Dr. Truong,

Thank you for submitting your manuscript to PLOS ONE. After careful consideration, we feel that it has merit but does not fully meet PLOS ONE’s publication criteria as it currently stands. Therefore, we invite you to submit a revised version of the manuscript that addresses the points raised during the review process.

We look forward to receiving your revised manuscript.

Kind regards,

Mingming Liu

Academic Editor

PLOS ONE

2.  We note you have not yet provided a protocols.io PDF version of your protocol and/or a protocols.io DOI. When you submit your revision, please provide a PDF version of your protocol as generated by protocols.io (the file will have the protocols.io logo in the upper right corner of the first page) as a Supporting Information file. The filename should be S1_file.pdf, and you should enter “S1 File” into the Description field. Any additional protocols should be numbered S2, S3, and so on. Please also follow the instructions for Supporting Information captions [https://journals.plos.org/plosone/s/supporting-information#loc-captions]. The title in the caption should read: “Step-by-step protocol, also available on protocols.io.”

Please assign your protocol a protocols.io DOI, if you have not already done so, and include the following line in the Materials and Methods section of your manuscript: “The protocol described in this peer-reviewed article is published on protocols.io (https://dx.doi.org/10.17504/protocols.io.[...]) and is included for printing purposes as S1 File.” You should also supply the DOI in the Protocols.io DOI field of the submission form when you submit your revision.

If you have not yet uploaded your protocol to protocols.io, you are invited to use the platform’s protocol entry service [https://www.protocols.io/we-enter-protocols] for doing so, at no charge. Through this service, the team at protocols.io will enter your protocol for you and format it in a way that takes advantage of the platform’s features. When submitting your protocol to the protocol entry service please include the customer code PLOS2022 in the Note field and indicate that your protocol is associated with a PLOS ONE Lab Protocol Submission. You should also include the title and manuscript number of your PLOS ONE submission.

“This work had been supported through the Innovation Fund by Microsoft Australia.”

“This study was funded by Innovation Grant (Microsoft Australia). The authors (LT, FA, DDS) were granted $35,000 AUD for the development of the automatic pipeline in Microsoft Azure server. The sponsor (Microsoft Australia) played no role in the study design, data analysis or preparation of the manuscript.”

6. Thank you for stating the following in your Competing Interests section: 

“No authors have competing interests”

7. We note that you have stated that you will provide repository information for your data at acceptance. Should your manuscript be accepted for publication, we will hold it until you provide the relevant accession numbers or DOIs necessary to access your data. If you wish to make changes to your Data Availability statement, please describe these changes in your cover letter and we will update your Data Availability statement to reflect the information you provide.

8. Please amend your manuscript to include your abstract after the title page.

Additional Editor Comments:

This paper presents an automatic data processing pipeline for nanopore sequence analysis using Microsoft Azure cloud computing service. In this process the claim is the automatic pipeline in the specific cloud environment which can be used for HLA genotype analysis. There is some merit in the proposed work but unfortunately the presentation lacks clarify and details. In addition to reviewers' comments, please include clarifications to the following concerns/questions.  1. Please consider including more references to your article. For instance, a related work section could be added to review some existing work in the literature.  2. Please discuss the limitation of your approach and any other features/settings that could be incorporated to further improve your pipeline. 3. Please present more details of the dataset used in your experiments.  

**Comments to the Author**

1. Does the manuscript report a protocol which is of utility to the research community and adds value to the published literature?

Reviewer #1: Yes

Reviewer #2: Yes

2. Has the protocol been described in sufficient detail?

Descriptions of methods and reagents contained in the step-by-step protocol should be reported in sufficient detail for another researcher to reproduce all experiments and analyses. The protocol should describe the appropriate controls, sample sizes and replication needed to ensure that the data are robust and reproducible.

Reviewer #1: Partly

Reviewer #2: Partly

3. Does the protocol describe a validated method?

Reviewer #1: Yes

Reviewer #2: No

4. If the manuscript contains new data, have the authors made this data fully available?

Reviewer #1: Yes

Reviewer #2: No

**5. Is the article presented in an intelligible fashion and written in standard English?**

Reviewer #1: Yes

Reviewer #2: Yes

6. Review Comments to the Author

Reviewer #1: This study was undertaken to address the problem of processing and analyzing data from the nanopore sequencing of HLA genes in a short period of time and with possibly lower cost. The study proposes a cloud-based solution that is stream-lined and increases the throughput and decrease the processing time and cost per sample genotyped.

The HLA data generated appear to be the same whether the investigators use the manually physical computing resources, or the cloud-based pipeline generated. This is positive.

Other comments:

1. Is it appropriate to compare the cost per sample of using the cloud-based pipeline, which is the cost it takes to use all the various different computer pieces to that of the cost of a technician for the manual process? There is no information regarding the costs for buying and maintaining the computing infrastructure for the manual process that they describe, which is not insignificant, especially if an institution doesn’t have a hig- performance computing cluster for use by the lab interested in running the ONT pipeline. This infrastructure cost should be combined with the cost for the technician’s time to account for all resources needed.

2. Based on data in Table 4, the underlying work and testing for building this pipeline occurred 1 – 2 years ago. Has the system you configured and built in Azure using the specific GPUs and CPUs described here, held up over time? Meaning, if you were to analyze an ONT run today, would you still be utilizing this hardware set up? And, if you are utilizing the same compute times, how has the cost changed in that time frame?

3. One thing that is not addressed is the time and effort it took to build this automated system. So let’s say your lab has a competent bionformatician or IT support person who is currently running a similar pipeline “in-house” as you were doing, how much extra effort was spent to build out this cloud-based resource? Is this something that someone can do it a matter of a few days, weeks, months? What kind of expertise are required?

4. For labs that do not have a dedicated bioinformatician/IT resources to put this pipeline into place, is this resource that you’ve built available for others to spin up and use?

5. Did the authors meet any challenges from their respective institutions when it came to utilizing cloud-based resources for clinical data? In our experience, many institutions are not trusting of these platforms for non-research-based activities, including clinical HLA genotyping. If you did, can you comment on the driving forces behind accepting a cloud-based pipeline instead of your in-house pipeline? It is not uncommon for a reduction in cost to not overcome the security factors of keeping the data on-site.

The manuscript needs significant editing. Two examples:

1. Page 16: Final sentence of the Discussion section should read “clinical history or treatment was not collected for the scope of this study.”, not scoop of this study.

2. Page 16: End of the 1st paragraph of Discussion should be “utmost security” not “upmost security”

Reviewer #2: In this protocol, the authors propose a pipeline to analyze the nanopore sequential data automatically utilizing cloud computation service from Microsoft Azure to accelerate HLA genotyping service. However, there are some comments the author may consider to improve the quality of the protocol.

Introduction section:

1. “The computational power that is needed to analyze nanopore sequencing data in a timely-manner and to enable long-term storage has also outgrown …”

Here is a bit confusing. Please give one or two more sentences to explain how computational power enable the long-term storage.

2. As the authors mentioned, there are cloud services provided by Google, Amazon and Microsoft and “All cloud providers offer elasticity, convenience and scalability depending on specific demand of individual workflow.”

Why the Microsoft Azure is chosen for the protocol?

Methods and Results section:

1. “The individual FASTQ reads were then further filtered by size, a minimum length of 2 kb, and quality, minimum Q-score of 7 ….”

Are there any references indicating that the threshold of minimum length and minimum Q-score are reasonable in the HLA genotyping? These thresholds are quite critical in data preprocessing.

Identifying optimal virtual machine section:

1. In table 1, the comparison between different GPU VMs, only NC6 v3 is tested and other types of GPU VMs has not been used. Why the NC6 v3 is the best option here? For instance, what if the runtime of NC24 v3 is extremely fast (even if it is expensive)?

2. In table 1, the “finding” gives the decision whether the VM is selected. However, in VM selection, what metrics you consider mostly here? For instance, a calculation based on the cost and runtime, which helps to generate the final decision.

3. There problems in table 2 are similar with that in table 1. Why the L32 is selected since the runtime and cost are not optimal compared with other types of VMs?

Please give one or two sentences to clearly summarize that the VM is selection based on what consideration.

Genetic data output comparison section:

1. What is the unit for y-axis in figure 3?

2. The x-axis is not legible.

3. Ideally, the number of reads detected in each sample (y-axis) should be identical between Azure cloud analysis and manual analysis. However, from BC22 to BC26, the gap between the two analysis methods seems a bit large. Will it cause any consequences in the following HLA genotyping?

4. In figure 4, ensure the “none”s are with a same font size.

5. The pie charts in figure 4 looks similar, are the any metrics to evaluate the similarity between them? (not just mention the two pie charts are comparable)

7. PLOS authors have the option to publish the peer review history of their article (what does this mean?). If published, this will include your full peer review and any attached files.

Reviewer #1: **Yes: **Dimitri Monos Ph.D

Reviewer #2: **Yes: **Hongde Wu

---

## [Author Response · Author response to Decision Letter 0]

20 Oct 2022

To address the academic editor’s comments:

1. The manuscript has been reformatted as per the provided PLOS ONE style template.

2. The step-by-step protocol was uploaded to protocols.io DOI and the PDF was attached as S1_file.pdf file. In addition, the statement referring to protocols.io was also added to the Methods section in the manuscript.

3. The author-generated code and script was included in the manuscript in the intention of sharing without restrictions upon publication of this work. Additional code and commands were also included in the step-by-step protocol in protocols.io DOI.

4. The grant information in the “Funding Information” and “Financial Disclosure” sections was updated to be consistent.

5. The funding-related text was removed from the Acknowledgments section in the manuscript. Please update the statement “This study was funded by Innovation Grant (Microsoft Australia). The authors (LT, FA, DDS) were granted $35,000 AUD for the development of the automatic pipeline in Microsoft Azure server. The sponsor (Microsoft Australia) played no role in the study design, data analysis or preparation of the manuscript.” Into the Funding Statement section. 

6. The statement for Competing Interests section was updated to PLOS ONE’s guidelines and read as follows “The authors have declared that no competing interests exist.”

7. The dataset is currently published on Dyrad repository (https://doi.org/10.5061/dryad.x0k6djhp4) and fully accessible to reviewers.

8. The abstract was included in the manuscript after the title page.

To address additional comments from the editor:

1. More references to cloud computing application in clinical services were added to our discussion section (last paragraph on page 12 of the manuscript).

2. The limitations of this study was also acknowledged in the discussion section (first paragraph on page 14 of the manuscript).

3. The demographics and details of the data set in this study were included in the first paragraph on page 10 of the manuscript. 

To address reviewer #1’s comments:

1. The rationale behind cost estimation exercise was elucidated on page 8 right after Table 3.

2. The gap between validation run and implementation was explained in the discussion section, page 14 of the manuscript. 

3. The lack in details on the effort and timeline to build the pipeline was clarified in the discussion section on page 14.

4. The dependency on dedicated IT support was listed as a major limitation of the study. 

5. The common concern with the usage of cloud server regarding security and patient’s sensitive information was addressed in the discussion on page 15.

6. The two spelling errors had been corrected. The manuscript had undergone critical review by all authors prior to the resubmission.

To address reviewer #2’s comments:

1. The remark on computational power and long-term storage was elaborated in the introduction section on page 2.

2. The rationale of choosing Azure server was included in the introduction on page 3 of the manuscript.

3. The reasoning for pre-determined quality threshold was elaborated in the methods and results section on page 4.

4. The findings from VM testing for GPU and CPU in Table 1 and Table 2, respectively, were explained in more details on page 5 and page 6.

5. The y-axis of Figure 3 was labelled to represent number of reads detected per sample after demultiplexing process.

6. The x-axis of Figure 3 was re-formatted in bigger front to be legible. 

7. The gap in read output between two pipelines and potential consequences were explained on page 10 of the manuscript.

8. Figure 4 was reformatted for consistency between two pie charts.

9. P-value was included to demonstrate the statistical significance between two datasets on page 10.

---

## [Decision Letter · Decision Letter 1]

21 Nov 2022

Nanopore sequencing data analysis using Microsoft Azure cloud computing service

PONE-D-22-07900R1

Dear Dr. Truong,

We’re pleased to inform you that your manuscript has been judged scientifically suitable for publication and will be formally accepted for publication once it meets all outstanding technical requirements.

Kind regards,

Mingming Liu

Academic Editor

PLOS ONE

Additional Editor Comments (optional):

Reviewers' comments:

Reviewer's Responses to Questions

**Comments to the Author**

1. Does the manuscript report a protocol which is of utility to the research community and adds value to the published literature?

Reviewer #1: Yes

Reviewer #2: Yes

2. Has the protocol been described in sufficient detail?

To answer this question, please click the link to protocols.io in the Materials and Methods section of the manuscript (if a link has been provided) or consult the step-by-step protocol in the Supporting Information files.

The step-by-step protocol should contain sufficient detail for another researcher to be able to reproduce all experiments and analyses.

Reviewer #1: Yes

Reviewer #2: Partly

3. Does the protocol describe a validated method?

Reviewer #1: Yes

Reviewer #2: No

4. If the manuscript contains new data, have the authors made this data fully available?

Reviewer #1: Yes

Reviewer #2: N/A

**5. Is the article presented in an intelligible fashion and written in standard English?**

Reviewer #1: Yes

Reviewer #2: Yes

6. Review Comments to the Author

Reviewer #1: The authors have adequately responded to the reviewers' comments. However, the manuscript needs carefully editing and reviewing as it still has a few spelling errors.

Reviewer #2: The authors have addressed all the issues mentioned in the comments. However, the text in Fig 3 and Fig 4 is still not legible. I suggest the authors fix this issues in the final version of paper.

7. PLOS authors have the option to publish the peer review history of their article (what does this mean?). If published, this will include your full peer review and any attached files.

Reviewer #1: No

Reviewer #2: **Yes: **Hongde Wu

---

## [Editor Report · Acceptance letter]

25 Nov 2022

PONE-D-22-07900R1 

Nanopore sequencing data analysis using Microsoft Azure cloud computing service 

Dear Dr. Truong:

I'm pleased to inform you that your manuscript has been deemed suitable for publication in PLOS ONE. Congratulations! Your manuscript is now with our production department. 

Kind regards, 

on behalf of

Dr. Mingming Liu 

Academic Editor

PLOS ONE